# Pairwise Adjusted Mutual Information

## Abstract

A well-known metric for quantifying the similarity between two clusterings is the adjusted mutual information. Compared to mutual information, a corrective term based on random permutations of the labels is introduced, preventing two clusterings being similar by chance. Unfortunately, this adjustment makes the metric computationally expensive. In this paper, we propose a novel adjustment based on pairwise label permutations instead of full label permutations. Specifically, we consider permutations where only two samples, selected uniformly at random, exchange their labels. We show that the corresponding adjusted metric, which can be expressed explicitly, behaves similarly to the standard adjusted mutual information for assessing the quality of a clustering, while having a much lower time complexity. Both metrics are compared in terms of quality and performance on experiments based on synthetic and real data.

## 1 Introduction

A well-known metric for quantifying the similarity between two clusterings of the same data is the adjusted mutual information [Nguyen *et al.*, 2009; Vinh *et al.*, 2010]. Compared to mutual information, this metric is *adjusted* against chance, meaning that the similarity cannot be due to randomness but only to the structure of the dataset, appearing in both clusterings. This is the reason why this metric is widely used in unsupervised learning, see [Zhang *et al.*, 2013; Thirion *et al.*, 2014; Taha and Hanbury, 2015; Yang *et al.*, 2016; Wang *et al.*, 2017] for various applications.

The standard way of adjusting mutual information against chance is through random label permutations of one of the clusterings [Vinh *et al.*, 2010]. Unfortunately, this adjustment makes the metric computationally expensive. Specifically, the time complexity of the metric is in $O(\max(k,l)n)$, where $k, l$ are the numbers of clusters in each clustering and $n$ is the number of samples [Romano *et al.*, 2014]. As a comparison, the time complexity of mutual information is equal to $O(kl)$ given the contingency matrix of the clusterings, i.e., the matrix counting the number of samples in each pair of clusters, one per clustering. The additional computational effort required by adjustment is significant as the number of samples $n$ is typically much larger than the numbers of clusters $k, l$.

In this paper, we propose a novel adjustment based on *pairwise* permutations. That is, we consider permutations where only two samples, selected uniformly at random, exchange their labels. We show that the corresponding adjusted metric, we refer to as *pairwise adjusted mutual information*, is as efficient as adjusted mutual information for assessing the quality of a clustering, with a much lower time complexity. In particular, the time complexity is the *same* as that of mutual information. The gain in complexity is significant, as the computation time is now independent of the number of samples $n$, given the contingency matrix.

The rest of the paper is organized as follows. We first provide the definition and key properties of adjusted mutual information in the general setting of information theory. We then introduce mutual information with pairwise adjustement and explain why the exact same properties are satisfied by

this new notion of adjusted mutual information. The application of both notions of adjustment to clustering, including the explicit expressions of the corresponding metrics, is presented in section 4. Experiments on both synthetic and real data are presented in section 5. Section 6 concludes the paper.

## 2 Adjusted mutual information

Let $P$ be the uniform probability measure on $\Omega = \{1, \dots, n\}$, for some positive integer $n$. Let $X, Y$ be random variables on the probability space $(\Omega, P)$. Without any loss of generality, we assume that $X$ and $Y$ are mapping from $\Omega$ to sets consisting of consecutive integers, starting from 1. Denoting by $H$ the entropy, the mutual information between $X$ and $Y$ is defined by [Cover and Thomas, 1991]:

$$I(X, Y) = H(X) + H(Y) - H(X, Y). \tag{1}$$

This is the information shared by $X$ and $Y$, which is equal to 0 if $X$ and $Y$ are independent. A distance between $X$ and $Y$ can then be defined by:

$$d(X, Y) = H(X, Y) - I(X, Y) = H(X|Y) + H(Y|X).$$

This distance, known as the variation of information, is a metric in the quotient space of random variables under the equivalence relation $X \sim Y$ if and only if there is some bijection $\varphi$ such that $X = \varphi(Y)$ [Meilă, 2003].

**Adjusted mutual information.** The adjusted mutual information between $X$ and $Y$, corresponding to the mutual information between $X$ and $Y$ *adjusted* against chance, is defined by:

$$\Delta I(X, Y) = I(X, Y) - \mathrm{E}(I(X, Y_\sigma)), \tag{2}$$

where $Y_\sigma$ is the random variable $Y \circ \sigma$, for any permutation $\sigma$ of $\{1, \dots, n\}$, and the expectation is taken over all permutations $\sigma$, chosen uniformly at random.

**Remark 1** (Normalization). *It is frequent to also normalize adjusted mutual information, so as to get a score between 0 and 1 [Vinh et al., 2010; Romano et al., 2014]. In this paper, we only focus on the adjustment step. Note that normalization can be equally applied to both considered notions of adjustment and thus be studied separately.*

We have the equivalent definition:

$$\Delta I(X, Y) = \mathrm{E}(H(X, Y_\sigma)) - H(X, Y),$$
$$= \frac{1}{2}(\mathrm{E}(d(X, Y_\sigma)) - d(X, Y)). \tag{3}$$

This equivalence follows from Proposition 1 and the fact that the definition is symmetric in $X$ and $Y$. All proofs are available in the supplementary material.

**Proposition 1.** *We have for any random variables $X$ and $Y$:*

$$H(X) = \mathrm{E}(H(X_\sigma)),$$
$$\mathrm{E}(H(X, Y_\sigma)) = \mathrm{E}(H(X_\sigma, Y)),$$
$$\mathrm{E}(I(X, Y_\sigma)) = \mathrm{E}(I(X_\sigma, Y)).$$

In view of (3), we expect $\Delta I(X, Y)$ to be positive if $X$ and $Y$ share information, as $X$ is expected to be closer to $Y$ (for the distance $d$) than to $Y_\sigma$, a randomized version of $Y$. There are specific cases where $\Delta I(X, Y) = 0$, as stated in Proposition 2; these cases will be interpreted in terms of clustering in section 4.

**Proposition 2.** *We have $\Delta I(X, Y) = 0$ whenever $Y$ (or $X$, by symmetry) is constant or equal to some permutation of $\{1, \dots, n\}$.*

**Adjusted entropy.** Observing that $H(X) = I(X, X)$, we define similarly the adjusted entropy of $X$ by:

$$\Delta H(X) = \Delta I(X, X) = H(X) - \mathrm{E}(I(X, X_\sigma)).$$

By (1), we get:

$$\Delta H(X) = \mathrm{E}(H(X, X_\sigma)) - H(X) = \frac{1}{2}\mathrm{E}(d(X, X_\sigma)). \tag{4}$$

Since $d$ is a metric, this shows that the adjusted entropy of $X$ is non-negative.

**Proposition 3.** *We have $\Delta H(X) = 0$ if and only if $X$ is constant or equal to some permutation of* $\{1, \ldots, n\}$.

Proposition 3 characterizes random variables with zero adjusted entropy. Again, this result will be interpreted in terms of clustering in section 4.

# 3 Pairwise adjustment

In this section, we introduce pairwise adjusted mutual information. The definition is the same as adjusted mutual information, except that the permutation $\sigma$ is now restricted to the set of pairwise permutations. Specifically, we consider permutations $\sigma$ for which there exists $i, j \in \{1, \ldots, n\}$ such that $\sigma(i) = j$ and $\sigma(j) = i$, whereas $\sigma(t) = t$ for all $t \neq i, j$. We consider the set of such permutations $\sigma$ where the samples $i, j$ are drawn uniformly at random in the set $\{1, \ldots, n\}$. We denote by $\sigma_{\mathrm{p}}$ such a random permutation. Observe that $\sigma_{\mathrm{p}}$ is the identity with probability $1/n$ (the probability that $i = j$).

**Pairwise adjusted mutual information.** We define the *pairwise adjusted mutual information* as:

$$\Delta_{\mathrm{p}} I(X, Y) = I(X, Y) - \mathrm{E}(I(X, Y_{\sigma_{\mathrm{p}}})).$$

This is exactly the same definition as the adjusted mutual information, except for the considered permutations $\sigma_{\mathrm{p}}$. It can be readily verified that the same properties apply, with the exact same proofs, a key property being that the random permutations $\sigma_{\mathrm{p}}$ and $\sigma_{\mathrm{p}}^{-1}$ have the same distributions. In particular, we have the analogue of (3):

$$\Delta_{\mathrm{p}} I(X, Y) = \mathrm{E}(H(X, Y_{\sigma_{\mathrm{p}}})) - H(X, Y),$$
$$= \frac{1}{2}(\mathrm{E}(d(X, Y_{\sigma_{\mathrm{p}}})) - d(X, Y)). \tag{5}$$

Moreover, $\Delta_{\mathrm{p}} I(X, Y) = 0$ whenever $X$ or $Y$ is constant or equal to some permutation of $\{1, \ldots, n\}$.

**Pairwise adjusted entropy.** We also define the *pairwise adjusted entropy* as:

$$\Delta_{\mathrm{p}} H(X) = \Delta_{\mathrm{p}} I(X, X) = H(X) - \mathrm{E}(I(X, X_{\sigma_{\mathrm{p}}})).$$

We have $\Delta_{\mathrm{p}} H(X) \geq 0$, with equality if and only if $X$ is constant or equal to some permutation of $\{1, \ldots, n\}$.

# 4 Application to clustering

Let $A = \{A_1, \ldots, A_k\}$ and $B = \{B_1, \ldots, B_l\}$ be two partitions of some finite set $\{1, \ldots, n\}$ into $k$ and $l$ clusters, respectively. Let $\Omega = \{1, \ldots, n\}$ and P be the uniform probability measure over $\Omega$. Consider the random variables $X$ and $Y$ defined on $(\Omega, \mathrm{P})$ by $X^{-1}(i) = A_i$ for all $i = 1, \ldots, k$ and $Y^{-1}(j) = B_j$ for all $j = 1, \ldots, l$. Note that $X(\omega)$ and $Y(\omega)$ can be interpreted as the *labels* $i$ and $j$ of sample $\omega$ in clusterings $A$ and $B$, for each $\omega \in \{1, \ldots, n\}$.

We denote by $a_i = |A_i|$ the size of cluster $A_i$, by $b_j = |B_j|$ the size of cluster $B_j$, and by $n_{ij} = |A_i \cap B_j|$ the number of samples both in cluster $A_i$ and cluster $B_j$, for all $i = 1, \ldots, k$ and $j = 1, \ldots, l$. The matrix $(n_{ij})_{1 \leq i \leq k, 1 \leq j \leq l}$ is known as the *contingency matrix*. Note that $a_i$ and $b_j$ are the sums of row $i$ and column $j$ of the contingency matrix, respectively.

**Adjusted mutual information.** A well-known metric for assessing the similarity $s(A, B)$ between clusterings $A$ and $B$ is the adjusted mutual information[1] $\Delta I(X, Y)$ between the corresponding random variables $X$ and $Y$. In words, this is the common information shared by clusterings $A$ and $B$ not due to randomness.

By Proposition 2, we have $s(A, B) = 0$ whenever clustering $A$ (or $B$, by symmetry) is trivial, that is, it consists of a single cluster or of $n$ clusters (one per sample). This is a key property, showing the interest of the adjustment.

---

[1]Recall that we don't normalize the metric, see Remark 1.

It is known that [Vinh *et al.*, 2010]:

$$s(A, B) = -\sum_{i=1}^{k}\sum_{j=1}^{l}\frac{n_{ij}}{n}\log\frac{n_{ij}}{n}$$

$$+ \sum_{i=1}^{k}\sum_{j=1}^{l}\sum_{c=(a_i+b_j-n)^+}^{\min(a_i,b_j)}\frac{a_i!b_j!(n-a_i)!(n-b_j)!}{n!c!(a_i-c)!(b_j-c)!(n-a_i-b_j+c)!}\frac{c}{n}\log\frac{c}{n}, \tag{6}$$

with the notation $(\cdot)^+ = \max(\cdot, 0)$. The time complexity of this formula, which is dominated by the second term, is in $O(\max(k,l)n)$ [Romano *et al.*, 2014]. In particular, it is linear in the number of samples $n$.

Interestingly, we can similarly assess the quantity of information $q(A)$ contained in clustering $A$ through the adjusted entropy $\Delta H(X)$ of the corresponding random variable $X$. This is the information contained in $A$ not due to randomness. We have $q(A) \geq 0$ and, by Proposition 3, $q(A) = 0$ if and only if clustering $A$ is trivial, that is, it consists of a single cluster or of $n$ clusters (one per sample).

Since $q(A) = s(A, A)$, it follows from (6) that:

$$q(A) = -\sum_{i=1}^{k}\frac{a_i}{n}\log\frac{a_i}{n} + \sum_{i,j=1}^{K}\sum_{c=(a_i+a_j-n)^+}^{\min(a_i,a_j)} + \frac{a_i!a_j!(n-a_i)!(n-a_j)!}{n!c!(a_i-c)!(a_j-c)!(n-a_i-a_j+k)!}\frac{c}{n}\log\frac{c}{n}.$$

The time complexity of this formula, also dominated by the second term, is in $O(kn)$. Again, this complexity is linear in the number of samples $n$.

**Pairwise adjusted mutual information.** The main contribution of the paper is the following new measure of similarity $s_{\mathrm{p}}(A, B)$ between clusterings $A$ and $B$, based on the pairwise adjusted mutual information $\Delta_{\mathrm{p}}I(X, Y)$ between the corresponding random variables $X$ and $Y$. We have an explicit expression for this similarity:

**Theorem 1.** *We have for any clusterings $A, B$:*

$$s_{\mathrm{p}}(A, B) = 2\sum_{i=1}^{k}\sum_{j=1}^{l}\frac{n_{ij}(n-a_i-b_j+n_{ij})}{n^2}\left(\frac{n_{ij}}{n}\log\frac{n_{ij}}{n} - \frac{n_{ij}-1}{n}\log\frac{n_{ij}-1}{n}\right)$$

$$+ 2\sum_{i=1}^{k}\sum_{j=1}^{l}\frac{(a_i-n_{ij})(b_j-n_{ij})}{n^2}\left(\frac{n_{ij}}{n}\log\frac{n_{ij}}{n} - \frac{n_{ij}+1}{n}\log\frac{n_{ij}+1}{n}\right).$$

The time complexity of this formula is in $O(kl)$, like mutual information. It is independent of the number of samples $n$, given the contingency matrix. Corollary 1 shows that the time complexity reduces to $O(m)$, where $m$ is the number of non-zero entries of the contingency matrix, provided the latter is stored in sparse format.

**Corollary 1.** *We have for any clusterings $A, B$:*

$$s_{\mathrm{p}}(A, B) = 2\sum_{i,j:n_{ij}>0}\frac{n_{ij}(n-a_i-b_j+n_{ij})}{n^2}\left(\frac{n_{ij}}{n}\log\frac{n_{ij}}{n} - \frac{n_{ij}-1}{n}\log\frac{n_{ij}-1}{n}\right)$$

$$+ 2\sum_{i,j:n_{ij}>0}\frac{(a_i-n_{ij})(b_j-n_{ij})}{n^2}\left(\frac{n_{ij}}{n}\log\frac{n_{ij}}{n} - \frac{n_{ij}+1}{n}\log\frac{n_{ij}+1}{n} + \frac{1}{n}\log\frac{1}{n}\right)$$

$$- 2\left(n^2 - \sum_{i=1}^{k}a_i^2 - \sum_{j=1}^{l}b_i^2 + \sum_{i,j:n_{ij}>0}n_{ij}^2\right)\frac{1}{n}\log\frac{1}{n}.$$

Similarly, we can define the quantity of information $q_{\mathrm{p}}(A)$ in clustering $A$ through the pairwise adjusted entropy $\Delta_{\mathrm{p}}H(X)$ of the corresponding random variable $X$. Again, $q_{\mathrm{p}}(A) \geq 0$, with $q_{\mathrm{p}}(A) = 0$ if and only if clustering $A$ is trivial.

130 **Corollary 2.** *We have for any clustering A:*

$$q_{\mathrm{p}}(A) = 2 \sum_{i=1}^{k} \frac{a_i(n - a_i)}{n^2} \left( \frac{a_i}{n} \log \frac{a_i}{n} - \frac{a_i - 1}{n} \log \frac{a_i - 1}{n} - \frac{1}{n} \log \frac{1}{n} \right).$$

131 Note that the time complexity of this formula in $O(k)$. It only depends on the number of clusters $k$,
132 and not on the number of samples $n$.

## 5  Experiments

134 In this section, we compare both notions of adjusted mutual information through experiments
135 involving synthetic and real data. The experiments are run on a computer equipped with an AMD
136 Ryzen Threadripper 1950X 16-Core Processor and 32 GB of RAM, with a a Debian 10 OS. All codes
137 and datasets used in the experiments are available in the supplementary material.

138 **Synthetic data.**   We start with the simple case of $n = 100$ samples with clusters of even sizes,
139 consisting of consecutive samples. Specifically, we consider the set of clusterings $A^{(s)}$, consisting of
140 clusters of size $s$ (except possibly the last one), for $s = 1, 2, \ldots, 100$. In particular, both $A^{(1)}$ and
141 $A^{(100)}$ are trivial clusterings while $A^{(5)}$ consists of 20 clusters of size 5.

142 Figure 1 gives the similarity between clusterings $A^{(10)}$ and $A^{(s)}$ with respect to $s$ in terms of adjusted
143 mutual information, for both notions of adjustment, i.e., $s(A^{(10)}, A^{(s)})$ and $s_{\mathrm{p}}(A^{(10)}, A^{(s)})$. We
144 observe very close behaviors, suggesting that both notions of adjustment tend to capture the same
145 patterns in the clusterings. Note that the maximum similarity is attained for $s = 10$ in both cases, as
146 expected. The similarity is equal to 0 for $s \in \{1, 100\}$ for both cases, in agreement with Proposition
147 2. We also observe local peaks at $s = 20, 30, \ldots, 90$, which can be interpreted by the fact that
148 clustering $A^{(10)}$ is a refinement of clustering $A^{(s)}$ for these values of $s$; similarly, the local peak at
149 $s = 5$ may be interpreted by the fact that clustering $A^{(5)}$ is a refinement of clustering $A^{(10)}$. The
Spearman correlation between both metrics over all values of $s$ is equal to 0.99.

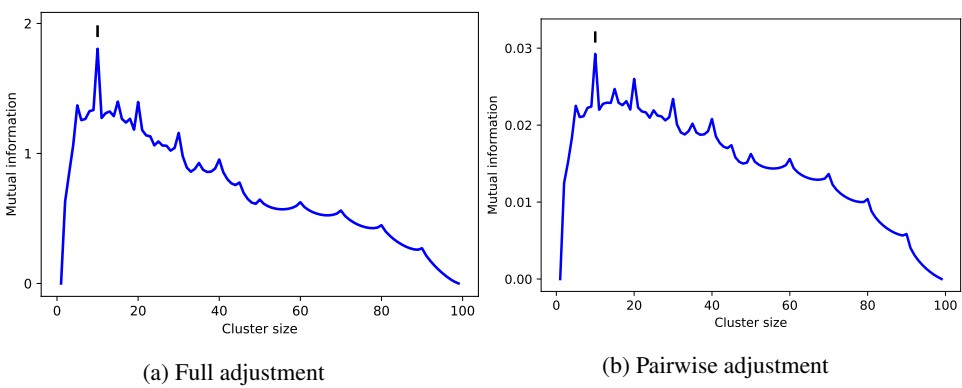

(a) Full adjustment                      (b) Pairwise adjustment

Figure 1: Comparison of metrics on synthetic data ($n = 100$).

150

151 We now consider random clusterings. Specifically, we assign $n$ samples to $k$ clusters independently
152 at random, according to some probability distribution $p = (p_1, \ldots, p_k)$, which is itself drawn at
153 random[2]. Consider three such random clusterings $A$, $B$, $C$ (with the same parameters $n$ and $k$, but
154 different probability distributions $p$). We would like to know whether $A$ is "closer" to $B$ or to $C$. In
155 particular, we are interested in testing whether both notions of adjusted mutual information give the
156 same ordering in the sense that:

$$(s(A, B) - s(A, C))(s_{\mathrm{p}}(A, B) - s_{\mathrm{p}}(A, C)) \geq 0. \tag{7}$$

---

[2]Namely, $p \propto U$ where $U = (U_1, \ldots, U_k)$ is a vector of $k$ i.i.d. random variables uniformly distributed over $[0, 1]$.

We compute the average precision score (fraction of triplets $A, B, C$ for which (7) is true) over $1\,000$ independent samples of $A, B, C$, for different values of $n$ and $k$. We repeat the experiment 100 times to get the mean and standard deviation. The results are given in Table 1. We observe a very high precision score, always higher than $93\%$, showing that both notions of adjusted mutual information tend to give the same ordering of these random clusterings.

| $n$ | $k$ | Precision score |
|------|------|------|
| 100 | 2 | $0.972 \pm 0.004$ |
| 100 | 5 | $0.952 \pm 0.007$ |
| 100 | 10 | $0.943 \pm 0.006$ |
| 100 | 20 | $0.955 \pm 0.008$ |
| 500 | 20 | $0.936 \pm 0.007$ |
| 1000 | 20 | $0.933 \pm 0.006$ |
| 1000 | 50 | $0.949 \pm 0.008$ |

Table 1: Precision score (mean $\pm$ standard deviation)

For the performance gain, we compare the computation times of both versions of adjusted mutual information for the similarity between clusterings $A$ and $B$, where $A$ consists of $k = 10$ clusters of same size and $B$ is a random clustering, drawn as in the previous experiment. Both versions of adjusted mutual information are coded in Python, with the standard version imported from scikit-learn. Figure 2 shows the computation time when the number of samples $n$ grows from $10^2$ to $10^7$. The performance gain brought by pairwise adjustement is significant. In particular, the computation time becomes independent of the number of samples.

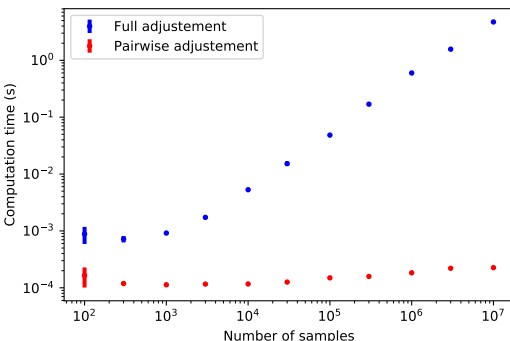

Figure 2: Computation time with respect to $n$ (mean $\pm$ standard deviation).

**Real data.**  We first consider the 79 datasets of the benchmark suite [Gagolewski, 2020][3]. We apply to each dataset each of the following clustering algorithms:

- $k$-means
- Affinity propagation
- Mean shift
- Spectral clustering
- Ward
- Agglomerative clustering
- DBSCAN
- OPTICS
- Birch
- Gaussian Mixture

---

[3]See https://github.com/gagolews/clustering_benchmarks_v1

We use the scikit-learn[4] implementation of these algorithms, with the corresponding default parameters[5]. We get 10 clusterings per dataset. The quality of each clustering is assessed through the similarity with the available ground-truth labels, using adjusted mutual information with either full adjustment or pairwise adjustment. We then compute the Spearman correlation of the corresponding similarities, a value of 1 meaning the exact same ordering of the 10 clusterings with full adjustment and pairwise adjustment. The results are shown in Figure 3, together with the speed-up in computation time due to pairwise adjustment. In both cases, the 79 datasets are ordered by the number of samples, ranging from 105 to 105 600 [Gagolewski, 2020].

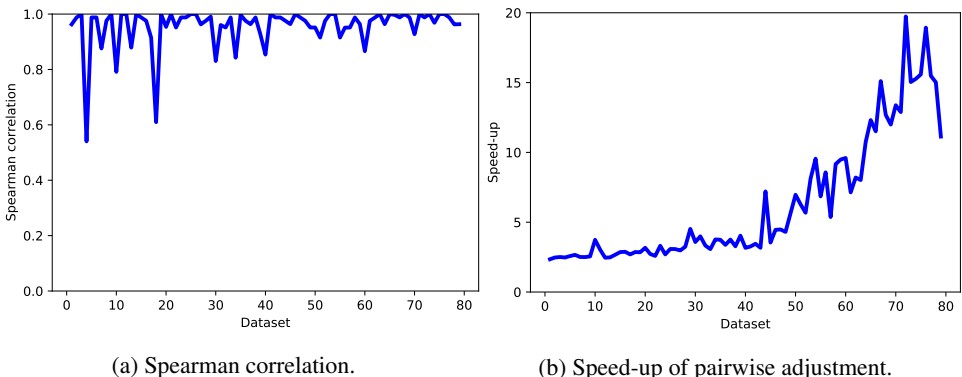

(a) Spearman correlation.                (b) Speed-up of pairwise adjustment.

Figure 3: Comparison of metrics on the Gagolewski benchmark.

We first observe that the correlation is very high, suggesting again that both notions of adjusted mutual information tend to provide the same results. For 65 datasets among 79, the Spearman correlation is higher than 95%. As for the computation time, we observe a significant performance gain, by one order of magnitude for the largest datasets.

We have conducted the same experiments with OpenML [Vanschoren *et al.*, 2013][6]. We selected all datasets with at least 1,000 but no more than 50,000 samples, at most 100 features (all numerical), no missing data and ground-truth labels forming clusters of at least 5 samples on average. The results are shown in Figure for the resulting 34 datasets. Again, the datasets are ordered by the number of samples, here ranging from 1,188 to 45,918. The conclusions are similar. In particular, the Spearman correlation is higher than 95% for 30 datasets among 34, and the performance gain exceeds 25 for the largest datasets.

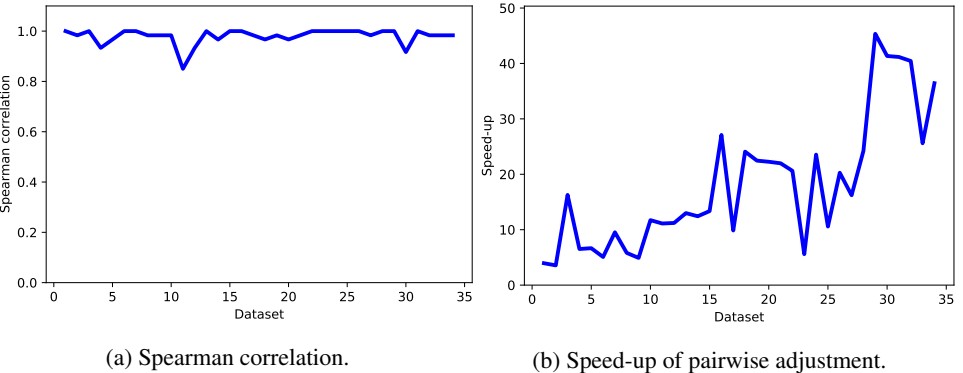

(a) Spearman correlation.                (b) Speed-up of pairwise adjustment.

Figure 4: Comparison of metrics on OpenML datasets.

---

[4]https://scikit-learn.org/

[5]Dimension reduction is applied to the MNIST datasets, consisting of 70 000 images of size $28 \times 28$ each, see the supplementary material for details.

[6]https://www.openml.org

## 6 Conclusion

We have proposed another way of adjusting mutual information against chance, through pairwise label permutations. The novel metric, whose explicit expression is given in Theorem 1, has a much lower complexity than the usual adjusted mutual information. Interestingly, both metrics can also be used to assess the quantity of information contained in a clustering, which the common property of being equal to 0 if and only if the clustering is trivial, as stated in Proposition 3; again, the pairwise adjusted entropy, given in Corollary 2, has a much lower complexity. Experiments on synthetic and real data show that pairwise adjusted mutual information tends to provide the same results as the usual adjusted mutual information for comparing clusterings, while involving much less computations.

For future work, we plan to extend this idea to other similarity metrics. While the practical interest is less obvious for the Adjusted Rand Index [Hubert and Arabie, 1985], due to the fact that the time complexity of this metric is already independent of the number of samples, it would be worth considering other versions of information theoretic measures, as those studied in [Romano *et al.*, 2016].

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
