# OpenReview forum: "Pairwise Adjusted Mutual Information"
_NeurIPS.cc/2021/Conference — NeurIPS 2021 Submitted_

### Official Review · Reviewer_xPxc · 2021-07-12

**Rating:** 5
**Confidence:** 4

**Summary:**

In this paper authors propose an efficient version of adjusted mutual information, termed pairwise adjusted mutual information, for cluster similarity evaluation. The topic is attractive and the results on both synthetic and real-world datasets demonstrate the effectiveness of the algorithm. However, the expression of the paper needs to be finely improved. Also, the depth of the theoretical analysis needs to be further improved. More analysis should be given.

**Limitations And Societal Impact:**

Please check the main review part.

**Main Review:**

There are grammar mistakes in the paper. It makes the paper relatively harder to understand. Here are some examples.
" Compared to mutual information, this metric is adjusted against chance, meaning that the similarity cannot be due to randomness but only to the structure of the dataset, appearing in both clusterings."
"As a comparison, the time complexity of mutual information is equal to O(kl) given the  contingency matrix of the clusterings, i.e., the matrix counting the number of samples in each pair of clusters, one per clustering."
"The gain in complexity is significant", do you mean the complexity decrease is significant?
To name just a few. The paper should be carefully polished before publishing.

The theoretical analysis of the paper is not thorough enough.
What is the theoretical relationship the adjusted mutual information and pairwise adjusted mutual information. What is the influence of introducing the restricted constraint? What is the gap between the two criteria? Similar conclusion of the proposed  pairwise adjusted mutual information is expected? How many pairwise permutations should be made to achieve good performance?

To the experimental part
More comprehensive experiments should be made to evaluate the stability of the proposed criterion.

**Time Spent Reviewing:**

4 hours

---

> ### Author Response · Authors · 2021-08-10
> **Number of pairwise permutations**
>
> > How many pairwise permutations should be made to achieve good performance?
>
> This question does not make sense. There is a single pairwise permutation, chosen uniformly at random. This is the basic idea of the paper, which has not even be understood by the reviewer.

---

> > ### Comment · Reviewer_xPxc · 2021-09-02
> > **Pairwise permutation**
> >
> > Many important training details are missing.
> > What is training iteration number, if the algorithm is stable in different iterations?
> > What is the setting of the Pairwise Adjusted Mutual Information in each iteration is not clear to me.

---

> > > ### Author Response · Authors · 2021-09-03
> > > **Pairwise permutation**
> > >
> > > There are no details about training because there is no training...
> > > The paper is about a metric, whose explicit expression is given in Theorem 1.
> > > Your question about iterations / stability does not make sense.

---

> > > > ### Comment · Reviewer_xPxc · 2021-09-03
> > > > **Metric**
> > > >
> > > > If the metric can only be used for measuring the similarity statically, its practicability will decrease.
> > > > Also, the urgency on improving the efficiency of calculating the metric would also decrease.

---

> > > > > ### Author Response · Authors · 2021-09-03
> > > > > **Metric**
> > > > >
> > > > > The AMI is a standard metric for measuring similarity, cf.
> > > > > https://scikit-learn.org/stable/modules/generated/sklearn.metrics.adjusted_mutual_info_score.html
> > > > >
> > > > > I don't see your point about its lack of practicability.

---

### Official Review · Reviewer_tfHx · 2021-07-20

**Rating:** 3
**Confidence:** 5

**Summary:**

This paper provides a new adjustment for the normalized mutual information, based on pairwise permutations to make the computation more efficient.  It shows that the new index correlates well with the original index in different experiments, while providing significant speedup.


**Main Review:**

Where do you get definition 2? Similarly, I am not able to verify proposition 1. Can you show how these are derived from the permutations?

How do you account for having fixed-sized clusters which is the assumption in the original AMI paper?

The contingency matrix computation is linear in n, so any computation on top that is also linear in n will not add to the overall complexity of measuring agreement between clusterings. The reason for AMI being expensive is that it is not linear in n, it is factorial!

The main issue with NMI is that it is sensitive to the number of clusters, that is random clusters with more clusters will get a better agreement with the ground truth. This is one of the key reasons for the need for an adjusted index. In most experiments, the number of clusters is fixed which doesn't allow for testing this. The sizes of the clusters are also an important factor in the adjustment for chance, which is overlooked by assuming even size clusters. Overall the experiments are limited and not convincing. The theory is not well explained. The related work is non-present.



**Time Spent Reviewing:**

3

---

> ### Author Response · Authors · 2021-08-10
> **Fixed-sized clusters**
>
> > How do you account for having fixed-sized clusters which is the assumption in the original AMI paper?
>
> We do not need this assumption. Clusters can have any sizes.

---

> ### Author Response · Authors · 2021-08-10
> **Proofs**
>
> > Where do you get definition 2? Similarly, I am not able to verify proposition 1.
>
> Please check the supplementary material.

---

### Official Review · Reviewer_n7gR · 2021-08-08

**Rating:** 4
**Confidence:** 4

**Summary:**

This paper proposes a new efficient measure, called the pairwise adjusted mutual information, to quantify the similarity between two partitions of a dataset. The proposed measure is a simplified version of well known adjusted mutual information and hence it is easy to compute. In particular, the time complexity is independent of the number of data points, while that of the standard fully adjusted mutual information is linear with respect to the number of data points. The proposal is empirically shown to be more efficient than the adjusted mutual information.

**Limitations And Societal Impact:**

Although the authors argue that the limitation is described, it is vague and the limitation of the proposed approach is not properly discussed. For example, it is important to discuss when it fails in finding of better clustering and what is the negative impact of the proposal if such a situation occurs.


**Main Review:**

### Originality

The originality of this paper is just one idea: reducing the full adjustment of the mutual information into the pairwise adjustment. Although combining a pairwise permutation to the adjustment of mutual information seems to be a new idea, the technique of pairwise permutation itself is often used in the context of statistical testing (pairwise permutation test). Therefore, I feel that the overall originality is not high enough compared to the standard of NeurIPS.

### Quality

The quality of this paper is not high as there is no theoretical analysis of the proposed approach. Hence the effectiveness of the pairwise adjustment is unfortunately not clear. Although its effectiveness is empirically evaluated, it is not convincing. In experiments in real-world datasets, the Spearman correlation is observed to compare the pairwise adjustment and the full adjustment, while this is not enough to empirically examine the effectiveness of the proposal. For example, it would be interesting to see the correlation between the proposal and other well known clustering measures such as adjusted Rand index and normalized mutual information.

### Clarity

This paper is overall clearly written. However, there are many parts that describe already known results (e.g. "adjusted mutual information" part in Section 4) and the overall length of the paper is just over 7 pages (maximum is 9 pages), hence I believe that much more contents can be put in this paper to increase its quality.

### Significance

Although I understand that the proposed pairwise adjustment is much faster than the full adjustment, the full adjustment is still as fast as (or faster than) the clustering process itself as its complexity is linear to the number of data points. Together with the fact that the Spearman correlation can be small depending on datasets shown in Figure 3, in a practical situation, I think there is no positive reason to employ the proposed pairwise adjustment and one can simply use the full adjustment. Therefore the significance of this paper is not high.


**Time Spent Reviewing:**

3

---

### Official Review · Reviewer_bn5i · 2021-08-18

**Rating:** 5
**Confidence:** 3

**Summary:**

This paper studies an adaptation of a well-known form of metric used to measure the similarity between two clusterings. The adjusted mutual information (AMI) rates the mutual information of two clusterings by using the expected mutual information obtained by randomly permuting the labels as the "benchmark". AMI has high computational cost (linear in the size of dataset). This work then proposes a new way of adjusting the mutual information that is much easier to compute and behaves similar to AMI in practice.

**Limitations And Societal Impact:**

The authors have adequately addressed limitations and societal impact as far as I'm concerned.

**Main Review:**

Originality: Most of the techniques (especially those used in the theoretical arguments) are old in this paper. The main novelty lies in the authors' observations that AMI is developed by considering the mutual information by using fully random permutations.

Quality: The work has some interesting insights, but I'm not sure if this venue would be the best place for receiving it. The key insight that replacing full permutations with partial (pairwise here) permutations can lead to significant savings in computing time is cute and has values. It adds to our understanding of clustering metrics. However, the work done in this paper, both theoretically and empirically, does not fully justify the claim that it is the perfect substitute for AMI. Also, I'm not sure about how significant the improvement is in computing efficiency (see more detailed comments below).

Clarity: The paper is clearly written and explained the main discoveries well. However, I would have liked it more if the authors could elaborate a bit more on the (theoretical) connection between the new metric and original AMI, especially since there is much space left. Right now the main body only contains the proposed new metric and the derivation of the terms, which does not provide enough insights.

Significance: I'm not very confinced about the significance of the findings. The proposed metric design twiddles the original full permutation in a nice way. However, I have difficulty understanding whether there is any guarantee than connects this new metric to the original one, as taking the expectation based on pairwise permutations can be very different from full permutation. It is intriguing that they seem to have close performance in the numerical experiments. However, the methods used to measure the similarity between these two metrics are also limited (mostly precision score and correlation) and indirect. The claim is broad and the datasets already tested do not seem to be sufficient.

I'm also concerned about the assumption that the contingency matrix is known. It seems to me that given any two clustering, the time spent on constructing the two clustering and computing the contingency matrix (all $n_{ij} = |A_i \cup B_j|$'s) might be dominating, so even if computing the pairwise AMI is cheap, it might result in marginal gain in overall time complexity.

**Time Spent Reviewing:**

3 hrs

---

### Decision · Program_Chairs · 2021-09-27

**Decision:**

Reject

**Comment:**

The paper suggests a new method for comparing clusterings, namely pairwise adjusted mutual information, which is related to the previously defined “adjusted mutual information”, except that the averaging is done over permutations swapping only two elements.  This allows faster computation.

All reviewers seem to be more or less in favor of rejection.  While I do not agree with all reviewers comments (for example: I actually do think the paper is clearly written) I do agree that this paper offers a nice definition but with little significance in both theory and practice, especially in the context of a venue like NeurIPS.  One possible direction to strengthen this paper could be, for example, to give a theoretical explanation to the results of the synthetic experiments.